# CHEMICAL NAMES STANDARDIZATION USING NEURAL SEQUENCE TO SEQUENCE MODEL

## ABSTRACT

Chemical information extraction is to convert chemical knowledge in text into true chemical database, which is a text processing task heavily relying on chemical compound name identification and standardization. Once a systematic name for a chemical compound is given, it will naturally and much simply convert the name into the eventually required molecular formula. However, for many chemical substances, they have been shown in many other names besides their systematic names which poses a great challenge for this task. In this paper, we propose a framework to do the auto standardization from the non-systematic names to the corresponding systematic names by using the spelling error correction, byte pair encoding tokenization and neural sequence to sequence model. Our framework is trained end to end and is fully data-driven. Our standardization accuracy on the test dataset achieves 54.04% which has a great improvement compared to previous state-of-the-art result.

## 1 INTRODUCTION

There are more than 100 million named chemical substances in the world. In order to uniquely identify every chemical substance, there are elaborate rules for assigning names to them on the basis of their structures. These names are called systematic names. The rules for these names are defined by International Union of Pure and Applied Chemistry (IUPAC) (Favre & Powell, 2013).

However, besides the systematic name, there can be also many other names for a chemical substance due to many reasons. Firstly, many chemical are so much a part of our life that we know them by their familiar names which we call them common names or trivial names for the sake of simplicity. For example, sucrose is a kind of sugar which we are very familiar with. Its systematic name is much more complicated, which is *(2R,3R,4S,5S,6R)-2-[(2S,3S,4S,5R)-3,4-dihydroxy-2,5-bis(hydroxymethyl)oxolan-2-yl]oxy-6-(hydroxymethyl)oxane-3,4,5-triol*.

Secondly, in chemistry industry, especially in pharmaceutical industry, many producers always generate new names to a chemical substance in order to distinguish their products from those of their competitors. We call these kind of names proprietary names. The most famous example is Aspirin. Its systematic name is *2-Acetoxybenzoic acid*. So due to the history reasons and idiomatic usages, a chemical substance can have many other names.

Chemical information extraction is a research that extracts useful chemical knowledge in text and converts it into a database, which strongly relies on the unique standard chemical names. Nowadays, there are many chemical databases such as PubChem and SciFinder, which are designed to store chemical information including chemical names, chemical structures, molecular formulas and other relevant information. For these databases, it is still an ongoing work to extract chemical information from chemical papers to update the databases. If all the chemical substances are expressed by the systematic names, it is easy to generate other information. For example, we can nearly perfectly convert the systematic name to other representations such as Simplified Molecular-Input Line-Entry System (SMILES) (Weininger, 1988) and International Chemical Identifier (InCHI) (Mcnaught, 2006) and then generate the structural formulas. Some online systems are already well developed for converting automatically systematic names to SMILES string with a very high precision such as Open Parser for Systematic IUPAC Nomenclature (OPSIN) (Lowe et al., 2011) developed by

Table 1: Examples of different types of error

| Error type | Non-systematic name | Systematic name |
|---|---|---|
| Spelling error | *benzoil chloride* | *benzoyl chloride* |
| | *1,3-benzoxazoole* | *1,3-benzoxazole* |
| Ordering error | *benzene, 1,4-dibromo-2-methyl* | *1,4-dibromo-2-methylbenzene* |
| | *4-pyrimidinecarbaldehyde* | *pyrimidine-4-carbaldehyde* |
| Common name error | *adenine* | *9H-purin-6-amine* |
| | *Aspirin* | *2-Acetoxybenzoic acid* |
| Synonym error | *sodium butoxide* | *sodium;butan-1-olate* |
| | *2-ethylfuroate* | *ethyl furan-2-carboxylate* |
| Mixed of synonym error | *2-hydroxy-8-iodonaphthalene* | *8-iodonaphthalen-2-ol* |
| and ordering error | *3-amino-1,2-benzisoxazole* | *1,2-benzoxazol-3-amine* |

University of Cambridge[1]. Unfortunately, nowadays a great number of the chemical substances are expressed by their non-systematic names in chemical papers, which increases significantly the difficulties for this task, so our work focuses on the standardization of non-systematic names. Examples of chemical information extraction are shown in Figure 1.

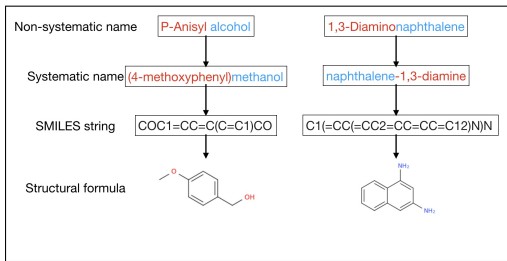

Figure 1: Examples of chemical information extraction (the parts in the same color means the same chemical constituent name)

In the following passage, we consider the differences between non-systematic names and systematic names as "error"[2]. In view of natural language processing, the error types of non-systematic names can be summarized by four types: 1. **Spelling error**. It means that non-systematic names just have slightly differences from systematic names in spelling; 2. **Ordering error**. It means that the groups in a non-systematic name are in wrong order; 3. **Common name error**. As mentioned above, many chemical substances have common names or proprietary names which look totally different from their systematic names; 4. **Synonym error**. It means that the words in the non-systematic names are different from those in the systematic names but they share the same root of word. In fact, it is the error type which happens most often. For example, *2-(Acetyloxy)benzoic Acid* has synonyms *Acetylsalicylic Acid* and *Acetysal* and these three words share the same root of word "Acety". Some examples of different types of errors are shown in Table 1. What is worth mentioning is that several types of error can appear at the same time in a single non-systematic name, especially for the ordering error and synonym error. The mixed types of error make this task very challenging.

Based on these four error types, we propose a framework to convert automatically the non-systematic names to systematic names. Our framework is structured as followed: 1. Spelling error correction. It aims to correct the spelling errors; 2. Byte pair encoding (BPE) tokenization. It aims to split a name into small parts; 3. Sequence to sequence model. It aims to fix all the remaining ordering errors, common name errors and synonym errors.

Actually, due to its great challenge, few work has been done on the chemical name standardization. To our best knowledge, Golebiewski et al. (2009) is the only work deserving a citation which

---

[1] Its precision reaches 99.8%.
[2] It does not mean that non-systematic names are wrong. It is just an expression for the differences.

developed an online system *ChemHits* to do the standardization basing on several transformation rules and the queries to online chemical databases. The work of Golebiewski et al. (2009) severely depends on chemical knowledge, limiting its application potential and effectiveness to some extent.

Differently, we adopt sequence to sequence model that has been widely used on neural machine translation. The reason why we apply the sequence to sequence model is that our task has some similarities with the machine translation problem. In machine translation, there are source language and target language which correspond to the non-systematic names and the systematic names in our task. Two different languages can be different in: 1. Vocabularies, which corresponds to the common name error and synonym error; 2. Word order, which corresponds to the ordering error. Our framework is trained end-to-end, fully data-driven and without using external chemical knowledge. With this approach, we achieve an accuracy of 54.04% in our test data set.

Our work will be done on a corpus containing chemical names extracted from report of Chemical Journals with High Impact factors (CJHIF)[3]. The corpus is collected and checked by paid manual work. It is a parallel corpus which includes non-systematic names and systematic names of chemical substances. In the following passage, we call a non-systematic name and the corresponding systematic name of a chemical substance data pair. In our corpus, there are 384816 data pairs. In Figure 2, we give an overview of the distribution of the Levenshtein distance between the non-systematic names and the systematic names to show how different the non-systematic names and the systematic names are. In the experiment, we use 80%, 19% and 1% data as training set, test set and development set respectively.

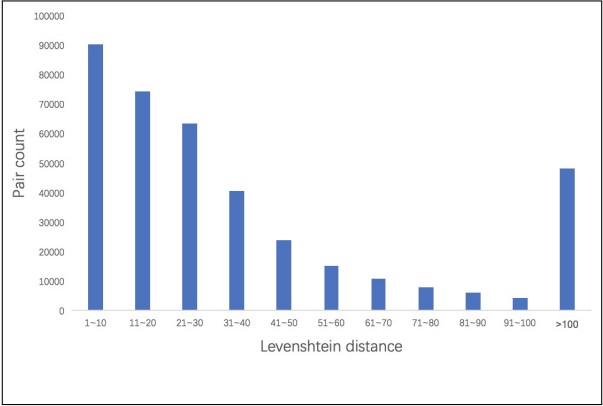

Figure 2: Distribution of the Levenshtein distance between non-systematic names and systematic names

## 2 PROPOSED FRAMEWORKS

Our framework consists of spelling error correction, byte pair encoding tokenization and sequence to sequence model which can be summarized in Figure 3.

### 2.1 SPELLING ERROR CORRECTION

In this part, we aim to correct the spelling errors. Given a name of a chemical substance, we can separate it into different elemental words by all the non-alphabet characters. For example, *2-(chloro-fluoro-methyl)-benzooxazole* can be separated into *chloro*, *fluoro*, *methyl* and *benzooxazole*. To correct the spelling error, firstly we set up two vocabularies from the dataset: vocabulary of the systematic elemental words and of the non-systematic elemental words. For the systematic elemental words, we just split all the systematic names to build the vocabulary. For the non-systematic elemental words, firstly we use all the non-systematic names to build an elemental vocabulary, and then we just keep the elemental words which appear many times in the non-systematic names but outside

---

[3]The full journal name list will be given later in supplementary material. Our corpus will be publicly released later.

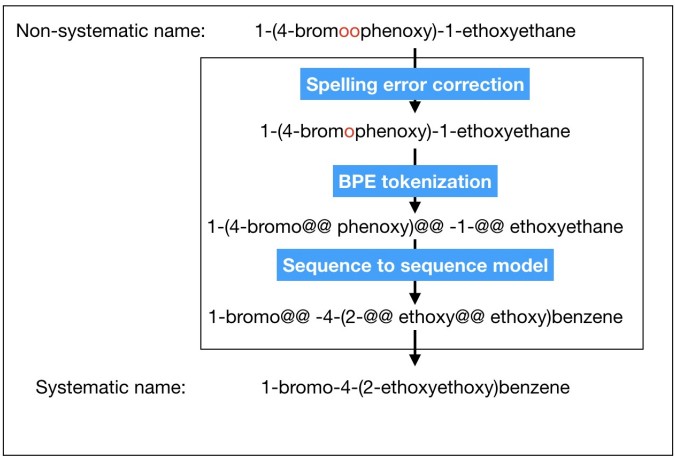

Figure 3: Illustration of the framework

the vocabulary of systematic elemental words and remove the rest. By this way, the vocabulary we build from the non-systematic names is the set of common names or synonyms. We then combine these two vocabularies together to get a final elemental vocabulary.

To do the correction search efficiently enough, we use BK-Tree (Burkhard & Keller, 1973) to structure the elemental vocabulary. BK-Tree is a tree structure which is widely used in spelling error correction. BK-Tree is defined in the following way. An arbitrary vocabulary item $a$ is selected as root node. The root node may have zero or more subtrees. The *k-th* subtree is recursively built of all vocabulary items $b$ such that $d(a, b) = k$ where $d(a, b)$ is the Levenshtein distance between $a$ and $b$. Given a word and a threshold, BK-Tree can return rapidly, if possible, the vocabulary item which have the smallest Levenshtein distance with the given word and the Levenshtein distance is smaller than the threshold by using the triangle rules: $|d(a, b) - d(b, c)| \leq d(a, c) \leq d(a, b) + d(b, c)$. By using the BK-Tree, we can correct the spelling error of non-systematic names. Another advantage of using BK-Tree is that it is easy to insert new training data which makes it scalable. An example of BK-Tree built from a part of our dataset is shown in Figure 4.

At this stage, given a name of a chemical substance, we firstly separate it into elemental words and then input the elemental words one by one to the BK-Tree. After the correction, we combine the elemental words to get the full name. In this step, a few non-systematic names can be directly corrected and some non-systematic names can be partially corrected. It is also helpful in the training of the sequence to sequence model because it can reduce the noise of elemental words.

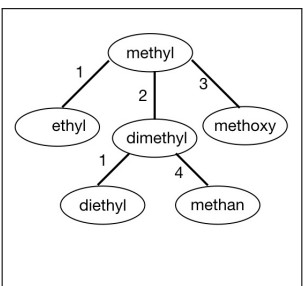

Figure 4: Example of BK-Tree built from a part of our dataset. Each node is an elemental word. The value on each edge is the Levenshtein distance between two nodes. All the nodes in the same subtree of a node have the same Levenshtein distance to this node. For instance, the Levenshtein distances from *dimethyl*, *diethyl*, *methan* to *methyl* are all 2.

Table 2: Examples of applying BPE to chemical names (subwords are separated by @@)

| Original name | Split name |
|---|---|
| *4-bromo-6-methoxyquinaldine* | *4-bromo@@ -6-methoxy@@ quinaldine* |
| *ethynyltris(propan-2-yl)silane* | *ethynyl@@ tris(propan-2-yl)@@ silane* |
| *methyltrioctylazanium bromide* | *methyl@@ trioctyl@@ azanium bromide* |

## 2.2 TOKENIZATION BY BYTE PAIR ENCODING

To apply the sequence-to-sequence model, firstly we need to tokenize all the chemical names. In this paper, we use Byte Pair Encoding (BPE) (Sennrich et al., 2015) to do the tokenization. Firstly, we initialize a symbol set by split all the names into characters. At this moment, the symbol set contains only the single characters. Then we iteratively count all symbol pairs and replace each occurrence of the most frequent pair (X, Y) with a new symbol XY and add it to the symbol set. Each merge operation produces a new symbol. The size of final symbol set is equal to the size of initial character, plus the number of merge operations. We then use the trained symbol vocabulary set to do the tokenization.

The reasons why we choose BPE are as follow: Firstly, it can deal with out-of-vocabulary problem because the vocabulary set generated by BPE contains the vocabularies at character level. Secondly, it can separate a name into meaningful subwords because it can find the small molecules which appear frequently in the corpus and tokenize a chemical name into the names of the small molecules. Some examples of applying BPE to the chemical name are shown in Table 2. After the tokenization, we can use the split pairs to train the sequence to sequence model.

## 2.3 SEQUENCE TO SEQUENCE MODEL

Sequence to sequence model (Sutskever et al., 2014) is widely used in machine translation. In this work, we adapted an existing implementation OpenNMT (Klein et al., 2017) with a few modifications. The sequence to sequence model consists of two recurrent neural networks (RNN) working together: (1) an encoder that gets the source sequences (here are the non-systematic names separated by BPE) and generates a context vector $H$, and (2) a decoder that uses this context vector to generate the target sequences (here are the corresponding systematic names). For the encoder, we use a multilayers bidirectional LSTM (BiLSTM) (Graves & Schmidhuber, 2005). BiLSTM consists of two LSTMs: one that processes the sequence forward and the other backward, with their forward and backward hidden states $\overrightarrow{h_t}$ and $\overleftarrow{h_t}$ at each time step. The hidden state at time step $t$ is just a concatenation of the two hidden states: $h_t = \{\overrightarrow{h_t}; \overleftarrow{h_t}\}$. At the final time step $T$ of the encoder, by combining all the hidden states, we get the context vector $H = \{h_1, ..., h_T\}$. For the decoder, it gives the probability of an output sequence $\hat{y} = \{\hat{y}_i\}$:

$$P(\hat{y}) = \prod_{t=1}^{M} p(\hat{y}_t | \{\hat{y}_{i<t}\})$$

and for a single token $\hat{y}_t$, the probability is calculated by

$$s_t = f(s_{t-1}, \hat{y}_{t-1})$$

$$\alpha_j = \frac{exp(score(s_t, h_j))}{\sum_{j'=1}^{T} exp(score(s_t, h_{j'}))}$$

$$c_t = \sum_j \alpha_j h_j$$

$$a_t = tanh(W_c[s_t; c_t])$$

$$p(\hat{y}_t | \{\hat{y}_{i<t}\}) = softmax(W_s a_t)$$

where $f$ is a multilayer LSTM; $s_t$ is the decoder's hidden state at time step $t$; and $W_c$, $W_s$ are learned weights. For the score function, we use the attention mechanism proposed by Luong et al. (2015):

$$score(s_t, h_j) = s_t^T W_a h_j$$

where $W_a$ are also learned weights.

## 3 EXPERIMENTS

### 3.1 TRAINING DETAILS

In our framework, at the spelling error correction stage, the only parameter is the threshold of the BK-Tree. In the experiment, we have tried several threshold values: 1, 2 and 3. At the BPE stage, the only parameter is the number of the merge operations. In the experiments, we have tried several values: 2500, 5000, 10000, 20000. For the sequence to sequence model, the dimensions of word embeddings and hidden states are both 500. The vocabulary size is equal to the number of basic characters plus the number of merge operations of BPE. The numbers of layers in encoder and decoder are both 2. Before training the sequence to sequence model, we also do the spelling error correction for the non-systematic names in the training data.

During training, all parameters of the sequence to sequence model are trained jointly using stochastic gradient descent (SGD). The loss function is a cross-entropy function, expressed as

$$L(y, \hat{y}) = -\sum_i y_i log(\hat{y}_i)$$

The loss was computed over an entire minibatch of the size 64 and then normalized. The weights are initialized using a random uniform distribution ranging from -0.1 to 0.1. The initial learning rate is 1.0 and the decay will be applied with the factor 0.5 every epoch after and including epoch 8 or when the perplexity does not decrease on the validation set. The drop out rate is 0.3 and we train the model for 15 epochs. We set the beam size to 5 for the decoding.

For comparison, we also do another experiment by replacing sequence to sequence model with Statistical Machine Translation (SMT) model. In this experiment, we use implemented Moses system Koehn et al. (2007). In the training, we limit the length of training sequences to 80 and apply the 3-grams language model by using KenLM (Heafield, 2011). The tokenization for the pairs we use is BPE with 5000 merge operations.

Besides the spelling error correction, data augmentation is another technique for the neural model learning to deal with the noisy data (in this case, the noise is the spelling error). For comparison, we also do the experiment of data augmentation. For every non-systematic name, we insert an error into it for the probability of 0.025. The error insertion has four types: we insert randomly a character in a random position; we randomly delete a character in a random position; we randomly exchange two characters; we randomly replace a character by another random character. The four insertion methods are applied in an equal probability.

### 3.2 RESULTS

In the experiment, we measure the standardization quality with accuracy and BLEU score (Papineni et al., 2002). Accuracy is calculated by the number of non-systematic names which are successfully standardized divided by the total number of non-systematic names. Note the accuracy that we adopt here is a very strong performance metric, as it equally means that the entire translated sentence is exactly matched for a machine translation task. The experiment results for different models on test dataset are shown in Table 3. We can see that the combination of spelling error correction, BPE tokenization and sequence to sequence model achieves the best performance. Our framework has a great improvement compared to the SMT model and the *ChemHits* system. The latter is slightly better than just applying spelling error correction. The results for different numbers of BPE merge operation are shown in Table 4. 5000 is the best value for this parameter. 0 means a character-level sequence to sequence model. The results show the usefulness of BPE. The results for different Levenshtein distance thresholds for the spelling error correction and the result of data augmentation are shown in Table 5. We can see that spelling error correction is indeed helpful for our framework.

Table 3: Results of different models on test dataset

| Models | Accuracy (%) | BLEU (%) |
|---|---|---|
| ChemHits (Golebiewski et al.) | 6.14 | - |
| Spelling error correction | 2.89 | - |
| Spelling error correction + SMT | 26.25 | 53.90 |
| **Spelling error correction + sequence to sequence** | **54.04** | **69.74** |

Table 4: Results for different numbers of BPE merge operation.

| Number of merge operation | Accuracy (%) | BLEU (%) |
|---|---|---|
| 20000 | 52.70 | 68.62 |
| 10000 | 53.55 | 69.22 |
| **5000** | **54.04** | **69.74** |
| 2500 | 53.90 | 70.19 |
| 0 | 23.60 | 64.25 |

Data augmentation also helps but does not perform as well as spelling error correction. Note that when the threshold is too large, the overcorrection might occur which reduces the standardization quality.

## 3.3 ANALYSIS

Some examples of the non-systematic names which are successfully standardized are shown in Table 6. These 4 examples show what the sequence to sequence model can do. In the first example, the parentheses in the non-systematic name are replaced by another parentheses and brackets. It means that the sequence to sequence model can fix also the non-alphabet spelling errors. The synonym error is also corrected: from *1-propanetriol* to *propane-1-thiol*. In the second example, the wrong order *ethane,1,2-dichloro* is corrected to *1,2-dichloroethane*. In the third example, the mixture of ordering error and synonym error are corrected. In the last example, *P-anise alcohol* is a proprietary name which looks like totally different from its systematic name but it is also successfully standardized.

To better illustrate how the sequence to sequence model works, here we give the visualization of attentions of an example, which is shown in Figure 5. The non-systematic name is *adenine,9-methyl- (7ci,8ci)* and the corresponding systematic name is *9-methyl-9H-purin-6-amine*. In the non-systematic name, *adenine* itself is also a chemical substance whose systematic name is *9H-purin-6-amine*. So it is a mixture of common name error and ordering error. From Figure 6, we can see that seq2seq model can find the relation between *adenine* and *9H-purin-6-amine* and can find the right place for *9-methyl*.

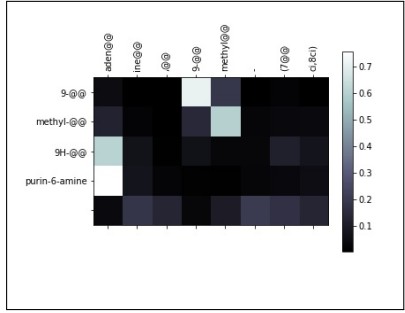

Figure 5: Visualization of attentions

Figure 6: Accuracy for different lengths

Table 5: Results of different Levenshtein distance thresholds for the spelling error correction. For each threshold, the first line is the accuracy after just doing the spelling error correction. The second line is the accuracy after processing the non-systematic name by our whole framework. The numbers of BPE merge operation are all 5000.

| Threshold | Accuracy (%) | BLEU (%) |
|---|---|---|
| Without spelling error | 0 | - |
| correction | 49.94 | 66.98 |
| **1** | **2.89** | - |
| | **54.04** | **69.74** |
| 2 | 2.51 | - |
| | 52.08 | 68.82 |
| 3 | 2.39 | - |
| | 53.84 | 69.60 |
| Data augmentation | 52.95 | 69.00 |

Table 6: Examples of the non-systematic names which are successfully standardized. For each example, the first line is the name before standardization and the second line is the name after standardization.

| | |
|---|---|
| Example 1 | *3-(dimethoxymethylsilyl)-1-propanetriol* |
| | *3-[dimethoxy(methyl)silyl]propane-1-thiol* |
| Example 2 | *ethane,1,2-dichloro* |
| | *1,2-dichloroethane* |
| Example 3 | *1-phenyl-3-methyl-4-benzoyl-1h-pyrazol-5(4h)-one* |
| | *4-benzoyl-3-methyl-1-phenyl-4,5-dihydro-1H-pyrazol-5-one* |
| Example 4 | *P-anise alcohol* |
| | *(4-methoxyphenyl)methanol* |

## 3.4 ERROR ANALYSIS

In this section, we will analyze the fail standardization attempts of our system. Firstly, we randomly select 100 samples of failed attempts and label their error types manually and carefully. The distribution over error types is shown in Table 7. We can see that synonym error is the most confusing error type and our system performs well at spelling error. As for the common error, since it is very hard to find a rule between an unseen common name and its systematic name, our system also perform poorly at this error type.

Among these 100 samples, there are 10 samples which are nearly correct (only one or two characters different from the systematic name), 7 examples are totally incorrect (none of the subwords of prediction match the systematic name) and the rest are partially correct. Some samples of failed attempts are shown in Table 8.

## 3.5 LIMITATIONS

We noticed that there are still nearly a half of the non-systematic names which are not successfully standardized. The accuracy for systematic names of different lengths are shown in Figure 6. We can see that our framework achieves the best performances for the systematic names of length between 20 and 40 while performing poorly for the systematic names of length bigger than 60 which account for 37% of our test dataset. Another limitation of our model is that we do not take into account chemical rules in our model. For this reason, a few names generated by our model disobey the chemical rules and at the tokenization stage, some subwords generated by BPE are not explicable as well.

Table 7: Distribution over error types of 100 failed attempts.

| Error types | Number of failed attempts |
| --- | --- |
| Synonym error | 59 |
| Common name error | 33 |
| Synonym error + ordering error | 7 |
| Spelling error | 1 |

Table 8: Examples of failed attempts. For each example, the first line is the name before standardization and the second line is the systematic name and the third line is the prediction of our model.

| | |
| --- | --- |
| Nearly correct | *5-bromo-2-(chlorosulfanyl)toluene* |
| | *4-bromo-2-methylbenzene-1-sulfonyl chloride* |
| | *5-bromo-2-methylbenzene-1-sulfonyl chloride* |
| Totally incorrect | *choline dicarbonate* |
| | *(2-hydroxyethyl)trimethylazanium hydrogen carbonate* |
| | *(carbamoylimino)urea* |
| Partially correct | *3,5 - dichloro - 4 - nitrobenzotrifluoride* |
| | *1,3-dichloro-2-iodo-5-(trifluoromethyl)benzene* |
| | *4,5-dichloro-2-nitro-1-(trifluoromethyl)benzene* |

## 4 CONCLUSION

In this work, we propose a framework to automatically convert non-systematic names to systematic names . Our framework consists of spelling error correction, byte pair encoding tokenization and sequence to sequence model. Our framework achieves an accuracy of 54.04% on our dataset, which is far better than previous rule based system (nine times of accuracy) and thus enables the related chemical information extraction into more practical use stage. The advantage of our framework is that it is trained end to end, fully data-driven and independent of external chemical knowledge. This work starts a brand new research line for the related chemical information extraction as to our best knowledge.

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
