# OpenReview forum: "CHEMICAL NAMES STANDARDIZATION USING NEURAL SEQUENCE TO SEQUENCE MODEL"
_ICLR.cc/2019/Conference_

### Official Review · AnonReviewer2 · 2018-10-25
**Solid paper with interesting application and dataset**

**Rating:** 7
**Confidence:** 3

**Review:**

This work presents a method to translate non-systematic names of chemical compounds into their systematic equivalents. Beyond that, a corpus of systematic and non-systematic chemical names is introduced that were extracted from chemistry and manually labelled.

The paper is well-structured and the authors introduce the problem setting very nicely to a machine learning audience, explain the challenges and motivate the architecture of their model. The model is a combination of tested approaches such as a spelling error correction, a byte pair encoding tokenizer and a sequence-to-sequence model consisting of (Bi)LSTMs and attention mechanisms.

The evaluation appears solid. The model achieves significantly improved results on the proposed corpus, even though compared to a underwhelming baseline. The analysis could be improved by showing and explaining some examples of failed translation attempts. It would also be interesting to see how the failed attempts are distributed over the error types (spelling, order, common name, synonym). The authors suggest a scenario where non-systematic names are converted and checked against a database of systematic names. For this, it would be interesting to know whether there are cases where a non-systematic name was translated into the wrong (but valid) systematic name.

Concluding, the paper presents an interesting application for machine translation, a new dataset and a method that successfully labels 50% of the given corpus.

Minor issue: The color scale of Fig. 5 is hard to recognize due to low contrast.

---

> ### Author Response · Authors · 2018-11-17
> **Thank you for your positive feedback and add the error analysis**
>
> Thank you for your positive feedback and insightful review. Because of the huge amount of data, we cannot check the error type of failed attempts one by one so we randomly select 100 fail attempts and label their error types manually and carefully. The result is reported in the section 3.4 of the updated version of our paper, in which some failed translation examples are also added. For the scenario, we admit that there could be the case where a non-systematic name is translated into the wrong (but valid) systematic name. For the preciseness, we delete this statement in the updated version of our paper.

---

### Official Review · AnonReviewer1 · 2018-10-31
**Important work in an underserved area but lacks ML contribution.**

**Rating:** 3
**Confidence:** 5

**Review:**

Pros: This seems like very competent and important work in an under-served area: Doing the mapping (or "entity linking") of chemical names to their standardized systematic forms. It's not my area, but I was frankly surprised when the paper said there was only one relevant prior piece of work, but having searched for a few minutes on Google Scholar, I'm at least inclined to believe that the authors are (approximately) right on that one. (This stands in stark contradistinction to the large quantity of biomedical entity recognition and linking work.) So, it's valuable to have work in this area, and the approach and application are sensible. In one sense, this gives the work significance and originality (as to domain). The paper is also clearly written, and certainly sufficiently accessible to an ML reader.

Cons: Unfortunately, though, I just don't think this qualifies for acceptance at ICLR. It's application of known techniques, and lacks any ML novelty or sufficient ML interest. It would only be appropriate for an "ML applications" track, which ICLR does not have. And while its performance is _way_ better than that of the only previous work on the topic that they know, accuracy of mapping non-systematic chemical terms (54.04%) is still low enough that this technique doesn't seem ready for prime time.

Other comments: In table 5, you show that a prior pipeline stage of spelling correction is definitely useful in your system (table 5). And yet, given the power of deep learning seq2seq transductions, and the potential to use them for spelling correction, one might wonder whether this prior step of spelling correction is really necessary. It might be interesting to explore further where it helps and whether the gains of spelling correction might be obtainable in other ways such as using data augmentation (such as spelling error insertion) in the seq2seq training data. The Golebiewski bib entry is lacking any information as to where it is published, which seems especially bad for the key citation to prior work of the whole paper. In general, the bibliography has issues: non-ASCII characters have been lost (you either need to LaTeX-escape them or to load a package like utf8, and capitalization of acronyms, etc. should be improved with curly braces.

---

> ### Author Response · Authors · 2018-11-17
> **Add the experiment of data augmentation**
>
> We appreciate a lot that you have affirmed that our work is one of the few works on this domain. Although the accuracy we achieve currently is still high enough, we have made an enormous progress compared to the previous method. As chemical name standardization is a very difficult problem, our work can serve as a strong baseline for the future research. For the spelling error correction, the idea of making the neural model to learn how to correct the spelling error is very interesting so we add an experiment using the data augmentation as you have mentioned. It achieves an accuracy of 52.95% and a BLEU score of 69.00. It is lower than the spelling error correction model (54.05%) but higher than the baseline (49.94%), which suggests that the data augmentation does work but does not perform as well as spelling error correction that we propose. We report this result (table 5) and the experiment details (section 3.1) in the updated version of our paper.

---

> > ### Comment · AnonReviewer1 · 2018-11-22
> > **Thanks!**
> >
> > Thanks for doing that extra experiment -- interesting! Gets you about 3/4 of the way from ~50 to ~54. I do think this work is worth publishing, maybe at an NLP venue, perhaps better even at a chemistry venue. However, I'm afraid I still don't think this is a great paper for ICLR.

---

> ### Author Response · Authors · 2018-11-23
> **The usefulness of our work**
>
> Please refer to our comments to AnonReviewer3 about our contribution and the role of this work.
> Here we want to add facts about the performance. According to our industrial partner,  the largest chemical database maybe only contains 2% chemical names appearing in all the chemical literature just because of the name obstacle. So, when an accuracy of about 60% is argued, we'd better look backwards. Only several percents of extraction accuracy has supported all the existing chemical database building for a broad range of practical application, while our system boosts nearly 10 times accuracy improvement compared to previous state-of-the-art. Just imagine how a new world we open for the community of CIP through this work.
> Please keep it in mind that even 6% extraction accuracy (ChemHits) can effectively and well serve all CIP work so far.

---

### Official Review · AnonReviewer3 · 2018-11-03
**Does not present new ideas**

**Rating:** 4
**Confidence:** 4

**Review:**

Name standardization is an important problem in Chemical Information Extraction. In the chemical literature, the same chemical could be referred to by many different names. This work focusses on standardizing non systematic names (for example Asprin to 2-Acetoxybenzoic acid).

The authors approach this as a machine translation problem. They create a parallel corpus of non-systematic and systematic names and build a seq2seq model for accomplishing this task. The authors report accuracy and BLEU score for the translation task.

My main concern with this work is novelty. It seems like this is a straightforward application of an existing model for this task. Also it is not clear why BLEU score is an appropriate metric for this task. Is it okay for downstream applications to have the systematic name partially right?

Overall, I think this paper does not present substantially new ideas to merit publication.

---

> ### Author Response · Authors · 2018-11-17
> **We still have important contribution**
>
> Thank you for your valuable comments. Although our work uses the existing models, our main contribution is that we combine these methods and create a pipeline to solve an import problem. We also prove that spelling error correction is useful for this application and we can extend this technique to all other domains which leverage the seq2seq model like neural machine translation. As you have mentioned, only a fully correct chemical name is useful for downstream application so we report also the accuracy of our method which is the key valuation metric. As for the BLEU score, it is widely used in machine translation. As mentioned in the paper, our task shares many similarities with machine translation task so BLEU score can provide another perspective to evaluate the result.

---

> ### Author Response · Authors · 2018-11-23
> **The importance of our work**
>
> We'd like to address the importance of our concerned task and how we make contribution other than model novelty.
>
> Chemical name identification and 3D-conversion is the core task for all chemical information processing (CIP), though from a view of NLP, it seems a routine information extraction that may be solved by various existing models. Without proper chemical names, all the rest CIP cannot be effectively done further. We already have mass data about chemistry including research papers and patents, from which we already have extracted a lot of chemical information databases. However, all the extraction is limited by the unsatisfactory accuracy on chemical names. So far, incomplete statistics show that the largest database maybe only contains 2% chemical names appearing in the literature, and the rest 98% that are supposed to be extracted cannot be exploited by CIP techniques just due to the name problem. The bottleneck of CIP is just located at its pipeline entrance, which is right what we do for in this work.
>
> We'd like to re-paraphrase our contribution as formulization or solution novelty from the non-trivality of our concerned task. As we know, our original aim in the task is to restore molecule structure just from a linear symbol sequence, chemical names which maybe have diverse spelling errors. It is surely uneasy from such a form by considering we have to let computer convert erring linear sequence into standard/exact 3D structure. Even though we re-formulize the most challenging part of the task into a chemical name correction subtask, it is still hard to solve. As no blanks are between chemical name stems and all the parts can be swap positions freely in different literature. We tried straightforward spell correction methods in NLP as the task looks like such a type, however, all methods including classical language model were shown to be a failure, for example, the most effectivel edit distance correction can only solve less than 2% cases. Machine translation solution come to us very late, whose effectiveness has been shown in this paper you are reading now.
>
> Overall, we contribute a novel ML solution for a bottleneck problem in CIP in this paper. Our components are not new maybe, but our solution especially for the task and our attempt that introduce these ML methods are new for sure.
>
> As we do not have enough space in the paper, we choose to submit the above comments here.

---

> > ### Comment · AnonReviewer3 · 2018-11-27
> > **Paper worth publishing**
> >
> > Thank you for highlighting the importance of your work. I do agree that this might be a paper worth publishing, but I am afraid that the paper may  not be a good fit for ICLR.

---

### Author Response · Authors · 2018-11-17
**Modification of our paper**

First of all, many thanks to all the valuable comments from the reviewers.

For the updated version of paper, we have modified the following things:
- Add the experiment of data augmentation (experiment details are in 3.1 and result is in table 5);
- Add a section 3.4 of error analysis;
- Delete the statement of application scenario.
- Improve the references

---

### Author Response · Authors · 2018-11-20
**About the novelty of our paper**

We acknowledge that our system has not introduced new models or components. However, this work still earns its merits by effectively formulizing such a chemical name processing problem into a machine translation task, which is not a trivial or straightforward solution from the problem form.  Our final and true purpose is to convert  chemimcal names with errors into its 3D forms, which is absolutely a non-trivial task for any known machine learning problem. Our main contribution is to effectively decompose such a problem and re-formulize the task into a machine translation one between error names and correct exact names. As this work is open, maybe it seems easy and staightforward, but it comes non-trivial problem and long-term unsuccessful attempts, and more importantly, it solves a long-term unsolved problem which is very needed in the current chemical inforrmation processing.
Overall, we report the first system for such a task that can give results with practical value.

---

### Meta-Review · Area_Chair1 · 2018-12-02
**Reject**

**Confidence:** 4
**Recommendation:** Reject

**Metareview:**

The area chair agrees with reviewer 1 and 2 that this paper does not have sufficient machine learning novelty for ICLR. This is competent work and the problem is interesting, but ICLR is not the right venue since the main contributions are on defining the task. All the models that are then applied are standard.